# Calponin-homology domain mediated bending of membrane-associated actin filaments

**Saravanan Palani[1,2], Sayantika Ghosh[1], Esther Ivorra-Molla[1], Scott Clarke[1], Andrejus Suchenko[1], Mohan K Balasubramanian[1]\*, Darius Vasco Köster[1]\***

[1]Centre for Mechanochemical Cell Biology and Warwick Medical School, Division of Biomedical Sciences, Coventry, United Kingdom; [2]Department of Biochemistry, Division of Biological Sciences, Indian Institute of Science, Bangalore, India

**Abstract** Actin filaments are central to numerous biological processes in all domains of life. Driven by the interplay with molecular motors, actin binding and actin modulating proteins, the actin cytoskeleton exhibits a variety of geometries. This includes structures with a curved geometry such as axon-stabilizing actin rings, actin cages around mitochondria and the cytokinetic actomyosin ring, which are generally assumed to be formed by short linear filaments held together by actin cross-linkers. However, whether individual actin filaments in these structures could be curved and how they may assume a curved geometry remains unknown. Here, we show that 'curly', a region from the IQGAP family of proteins from three different organisms, comprising the actin-binding calponin-homology domain and a C-terminal unstructured domain, stabilizes individual actin filaments in a curved geometry when anchored to lipid membranes. Although F-actin is semi-flexible with a persistence length of ~10 μm, binding of mobile curly within lipid membranes generates actin filament arcs and full rings of high curvature with radii below 1 μm. Higher rates of fully formed actin rings are observed in the presence of the actin-binding coiled-coil protein tropomyosin and when actin is directly polymerized on lipid membranes decorated with curly. Strikingly, curly induced actin filament rings contract upon the addition of muscle myosin II filaments and expression of curly in mammalian cells leads to highly curved actin structures in the cytoskeleton. Taken together, our work identifies a new mechanism to generate highly curved actin filaments, which opens a range of possibilities to control actin filament geometries, that can be used, for example, in designing synthetic cytoskeletal structures.

**\*For correspondence:**
m.k.balasubramanian@warwick.ac.uk (MKB);
D.Koester@warwick.ac.uk (DVK)

## Introduction

The IQGAP family of proteins plays a key role in actin cytoskeleton regulation including the assembly and function of the contractile actomyosin ring in budding and fission yeasts (*Briggs and Sacks, 2003*; *Eng et al., 1998*; *Epp and Chant, 1997*; *Lippincott and Li, 1998*; *Tebbs and Pollard, 2013*). To study the mechanism and role of actin binding by the fission yeast IQGAP (encoded by the *rng2* gene), we utilized a strategy to investigate its function when immobilized on supported lipid bilayers. We chose this approach, since during cytokinesis Rng2, which binds several actomyosin ring proteins, is tethered to the plasma membrane via Mid1, ensuring the formation and anchoring of the cytokinetic ring (*Laplante et al., 2016*; *Laporte et al., 2011*; *Padmanabhan et al., 2011*). We linked hexa-histidine tagged Rng2(1-189) to supported lipid bilayers containing nickel-chelating lipids (DOGS-NTA(Ni$^{2+}$)) and observed the binding of fluorescently labeled actin filaments using live total internal reflection fluorescence (TIRF) microscopy as described earlier (*Köster et al., 2016*; *Figure 1A*; *Figure 1—figure supplement 1A*). Remarkably, actin filaments landing onto His$_6$-Rng2 (1-189) decorated SLBs formed highly bent arcs and full rings with curvatures of $C_{curly} = 1.7 \pm 0.5$

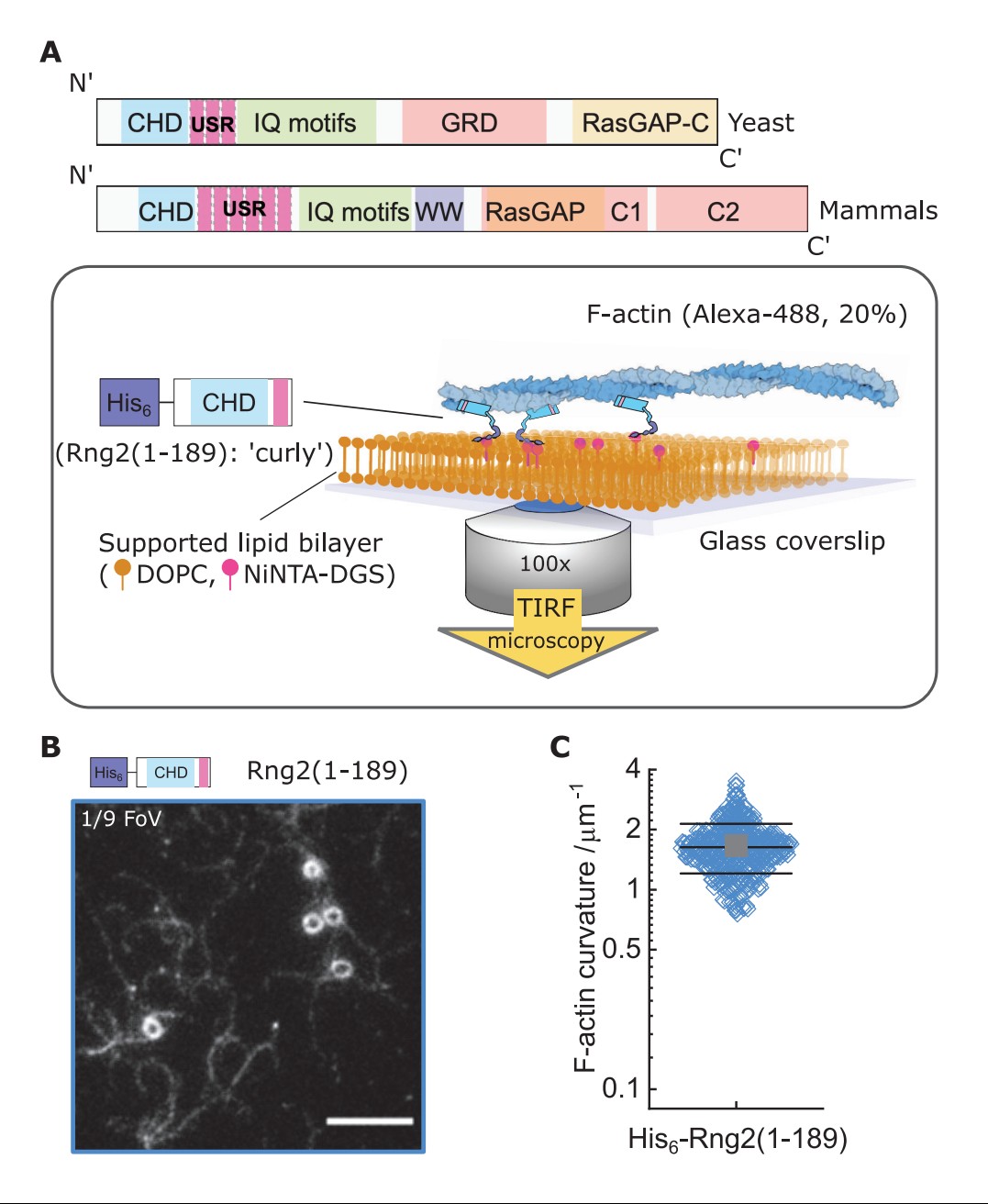

**Figure 1.** Formation of actin filament rings by membrane tethered curly (His$_6$-Rng2(1-189)). (**A**) Schematic representation of (top) the IQGAP proteins Rng2 (yeast, *S. pombe*) and IQGAP1 (mammals, *H. Sapiens*) and (bottom) the experimental setup used in this study; CHD - Calponin Homology Domain, USR – Unstructured Region, GRD - GAP Related Domain, RasGAP – Ras GTPase Activating Protein, WW – tryptophan containing protein domain. (**B**) TIRF microscopy image of actin filaments (Alexa488, $C_{actin}$ = 100 nM) bound to SLB tethered His$_6$-curly ($C_{curly}$ = 10 nM); shown is 1/9 field of view (FoV), scale bar 5 µm. (**C**) Curvature measurements of actin filament rings and curved segments; shown are the individual data points and their mean ± s.d.; N = 425 obtained from five field of views from each of four independent experiments.

The online version of this article includes the following figure supplement(s) for figure 1:

**Figure supplement 1.** Protein purification and measurements of actin rings induced by curly.

**Figure supplement 2.** Other actin binding proteins do not bend actin filaments.

$\mu m^{-1}$ (*Figure 1B,C*; *Figure 1—figure supplement 1D–G*; *Videos 1* and *2*). Binding of other membrane tethering actin binding proteins in the same geometry, such as the CHD of α-actinin or the actin binding domain of Ezrin, did not appreciably bend actin filaments (*Figure 1—figure supplement 2A–D*). Membrane anchored fimbrin has also been shown not to bend actin (*Murrell and Gardel, 2012*). To our knowledge, the bending of individual actin filaments into rings is an unprecedented phenomenon among the known actin binding proteins and we will refer in the following to His6-Rng2(1-189) as 'curly'.

To understand the mechanism leading to actin filament bending and ring formation by curly, we tested the role of curly anchoring to lipid membranes and its orientation. Curly mobility within planar lipid membranes was important for actin bending as glass adsorbed, immobilized $His_6$-Curly led to reduced actin binding and bending ($C_{glass} = 0.6 \pm 0.3$ $\mu m^{-1}$) (*Figure 2A*). However, placing the hexa-histidine tag to the C-terminus, Rng2(1-189)-$His_6$, did not affect actin filament bending ($C_{curly-his} = 1.5 \pm 0.5$ $\mu m^{-1}$) (*Figure 2B–D*, *Figure 2—figure supplement 1*).

Next, we studied whether actin bending by curly depended on the orientation of actin filaments by following the landing of already polymerized actin filaments decorated with labeled capping protein as a plus end marker (*Bieling et al., 2016*). We found that the bending was oriented anti-clockwise with respect to the plus end in all instances, wherein the plus end was clearly labeled to identify the orientation of filament bending (*Figure 3A,B*; *Figure 3—figure supplement 1A,B*). This was observed using both, the N-terminal and C-terminal hexa-histidine tagged curly, indicating that the internal sequence of the two actin binding sites within curly sets the chirality of actin bending and not the position of the membrane linker (*Figure 3A,B*; *Figure 3—figure supplement 1A,B*; *Videos 3* and *4*). Actin filaments appeared to bend concomitant with their landing on the supported lipid bilayer, which indicates that the bending did not require the full actin filament to be tethered to the SLB and underlined the earlier observation that the bending occurred locally.

To decouple the actin filament bending from the landing of actin filaments, we induced the polymerization of new actin filaments on planar lipid membranes by membrane tethered formin ($His_6$-SpCdc12(FH1-FH2)), profilin-actin and ATP in the presence of membrane tethered $His_6$-Curly. Strikingly, actin filaments displayed characteristic bending shortly after the onset of polymerization ($C_{formin, short} = 1.1 \pm 0.3$ $\mu m^{-1}$) and often grew into full rings ($C_{formin\ rings} = 1.7 \pm 0.4$ $\mu m^{-1}$) (*Figure 3C,D*; *Figure 3—figure supplement 2A–E*; *Video 5*). By contrast, polymerization of actin filaments along SLBs decorated with $His_{10}$-SNAP-EzrinABD did not result in the formation of arcs or rings, establishing that actin filament bending was due to curly and not due to formin (*Figure 3—figure supplement 2F,G*). These observations show that actin bending occurs continuously due to the binding of membrane tethered curly and did not require the cross-linking of adjacent ends of the same filament as was observed with the actin cross-linker anillin (*Kučera et al., 2020*). Importantly, the uni-directional bending supports the hypothesis that the binding site of curly with actin filaments defines an orientation with the propagation of an established curved trajectory.

Actin filaments forming the cytokinetic ring in *S. pombe* are tightly associated with tropomyosin (Cdc8). By contrast, the actin cross-linker fimbrin is present outside the cytokinetic ring region in Arp2/3 generated actin patches and prevents tropomyosin binding to these patches (*Skau and Kovar, 2010*). To determine whether the actin bending effect of curly is conserved in tropomyosin-wrapped actin filaments, we incubated actin filaments with tropomyosin before adding them to $His_6$-Curly containing SLBs. Strikingly, addition of tropomyosin to actin filaments increased the frequency of actin ring formation without affecting actin filament curvature, while actin filaments incubated with the actin cross-

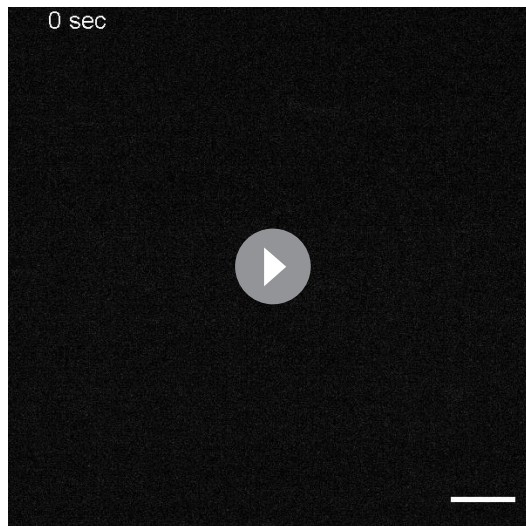

**Video 1.** TIRF microscopy image sequence of actin filaments (Alexa488) landing on $His_6$-curly decorated SLBs; scale bar: 5 μm.

https://elifesciences.org/articles/61078#video1

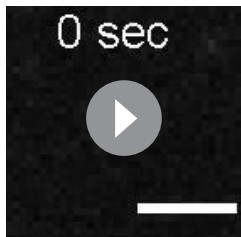

**Video 2.** TIRF microscopy image sequence of an individual actin filament (Alexa488) landing on His6-curly decorated SLBs and displaying multiple rounds of curling; scale bar: 1 μm.

https://elifesciences.org/articles/61078#video2

linker fimbrin displayed reduced bending and ring formation (*Figure 3—figure supplement 3*; *Video 6*). Thus, the tropomyosin Cdc8 and curly cooperate to enhance actin filament bending and ring formation.

Interestingly, we could observe that long actin filaments coated with tropomyosin would trace consecutive rings around the same center while landing on curly decorated lipid membranes. Subtraction of the image after completion of the first round of actin filament landing into a ring from the image after the second round revealed that the second ring occupied the interior space of the first ring. In line with that, a comparison of the intensity profiles perpendicular to the actin filament of the first and second round of ring formation revealed a widening of the profile toward the ring's interior (*Figure 3—figure supplement 4A,B*). A similar effect could be observed in examples of actin filaments polymerized by membrane tethered formin in the presence of membrane tethered curly (*Figure 3—figure supplement 4C,D*). This would suggest that curly can arrange long actin filaments into an inward-oriented spiral.

To test whether the curly induced actin rings can contract, we added rabbit skeletal muscle myosin II filaments and ATP to actin filaments bound to SLB tethered His$_6$-curly and followed actin filament dynamics over time. Shortly after myosin II addition, actin filaments (curved and straight ones) were propelled by myosin action leading to increased bending, rotation and finally to ring formation and contraction (*Figure 3E–G*, *Figure 3—figure supplement 5A–D*, *Video 7*). Interestingly, most actin rings displayed a counter-clockwise rotation (34/36 cases) and a slow contraction speed of $3 \pm 2$ nm s$^{-1}$ (*Figure 3F*; *Video 8*). The density of actin rings was strongly increased after myosin II addition (*Figure 3—figure supplement 5E*), indicating that actin sliding leads to more recruitment of curly. In line with this, actin filament rings displayed increased localization of fluorescently labeled curly after addition of myosin II filaments (*Figure 3—figure supplement 5F–H*). Despite reaching very high actin curvatures (up to 6.3 μm$^{-1}$) no breaking of actin filaments during the contraction process could be observed suggesting that binding of curly reduces the rigidity of actin filaments.

Since Rng2 belongs to the IQGAP protein family, we tested the N-terminal hexa-histidine tagged fragments of the IQGAP proteins Iqg1(1-330) (*S. cerevisiae*) and IQGAP1(1–678) (*H. sapiens*) and found that the bending of actin filaments was conserved ($C_{S.C.} = 1.1 \pm 0.4$ μm$^{-1}$; $C_{H.S.} = 1.0 \pm 0.2$ μm$^{-1}$) (*Figure 4A–C*; *Figure 4—figure supplement 1A*). Comparison of the available crystal structures of *H. sapiens* IQGAP1(28–190) with *S. pombe* Rng2(32-190) indicates high similarity between the two (*Figure 4—figure supplement 1B*).

Finally, to test the effect of curly on the actin cortex in cells, we expressed GFP -tagged curly in the mammalian cell lines HEK293T and RPE-1, which resulted in striking changes of the actin cortex architecture with the prominent occurrence of curved and ring-shaped actin filaments and bundles with curvatures of $C_{HEK293T} = 2.3 \pm 0.4$ μm$^{-1}$ and $C_{RPE-1} = 1.9 \pm 0.6$ μm$^{-1}$ (*Figure 4D–F*, *Figure 4—figure supplement 2A–D*, *Figure 4—figure supplement 3*). Co-expression with LifeAct-mCherry confirmed EGFP-Rng2(1-189) bound to actin filaments in cells (*Figure 4—figure supplement 2A,B*). To our surprise, addition of a CaaX domain to tether curly to cellular membranes did not enhance the actin bending effect but located curly mainly to structures resembling the endoplasmic reticulum (*Figure 4—figure supplement 4A*). The EGFP-Rng2(144-189) and EGFP-Rng2(144-189)-CaaX constructs showed only cytoplasmic localization (*Figure 4—figure supplement 4B,C*). To test whether the actin bending effect is conserved in the mammalian full-length protein, we imaged HEK293T cells expressing full-length EGFP-IQGAP1 and LifeAct-mCherry using lattice light sheet microscopy and indeed observed curved and ring-shaped actin structures inside the cell as well at its surface (*Figure 4—figure supplement 5*; *Video 9*). These experiments established that curly could instructively reorganize actin filaments into curved structures and rings and that this capacity is conserved in full-length IQGAP1.

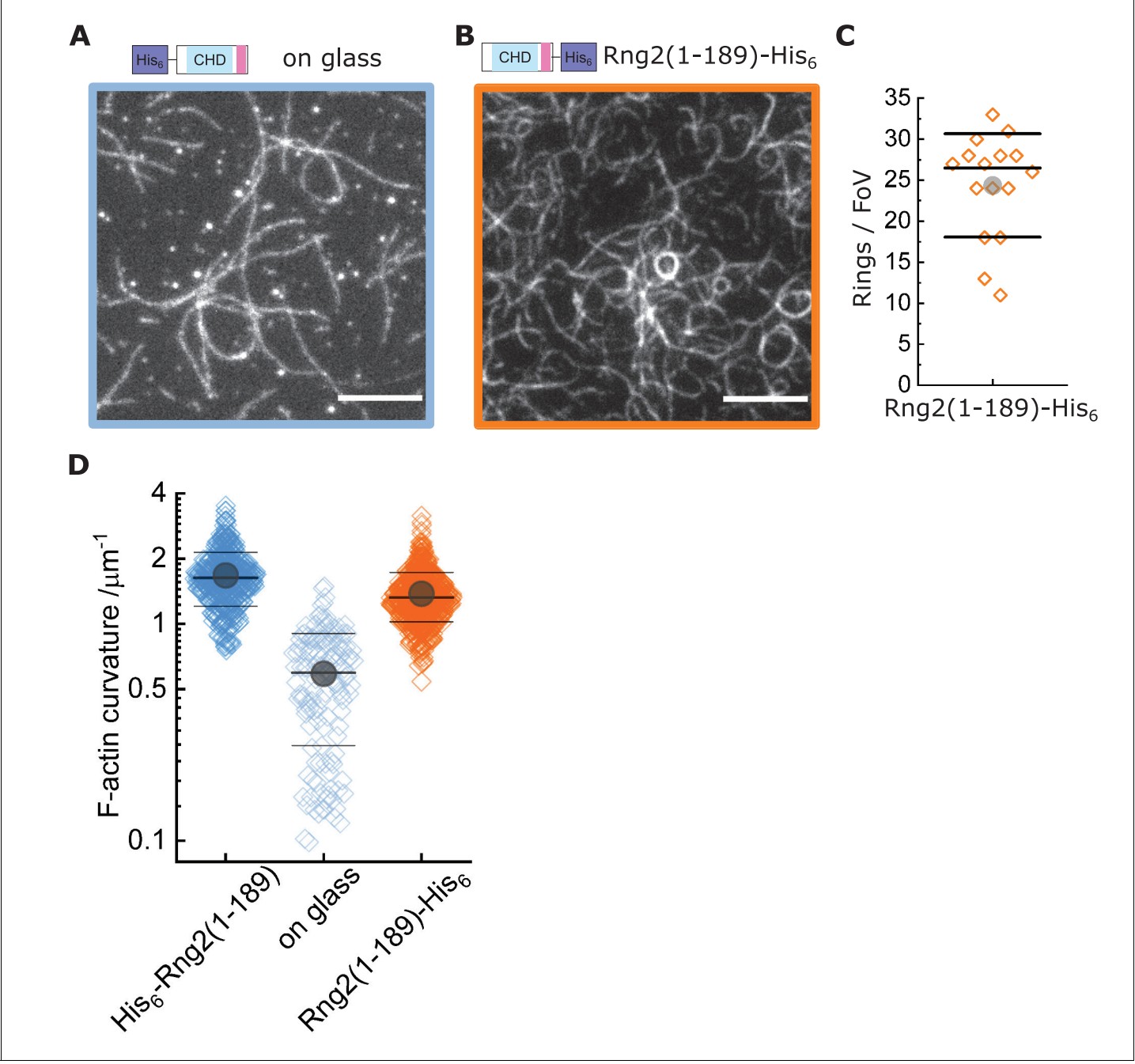

**Figure 2.** Actin bending is independent of curly orientation but requires lipid membrane tethering. TIRF microscopy images of actin filaments (Alexa488, $C_{actin}$ = 100 nM) bound to. (**A**) Glass adsorbed His$_6$-curly ($C_{curly}$ = 10 nM); N = 138 from nine field of views from each of three independent experiments. (**B**) SLB bound Rng2(1-189)-His$_6$($C_{Curly-His}$ = 10 nM); N = 658 from 13 field of views from each of four experiments; images show 1/9 field of view (FoV); scale bars: 5 μm. (**C**) Number of full actin rings per field of view induced by SLB tethered Rng2(1-189)-His6; N = 16 FoVs from three independent samples. (**D**) Comparison of curvature measurements of actin filament rings and curved segments; diamonds represent individual measurements, lines the median ± standard deviation and the circle the mean value.

The online version of this article includes the following figure supplement(s) for figure 2:

**Figure supplement 1,** Statistical analysis of data in Figure 2.

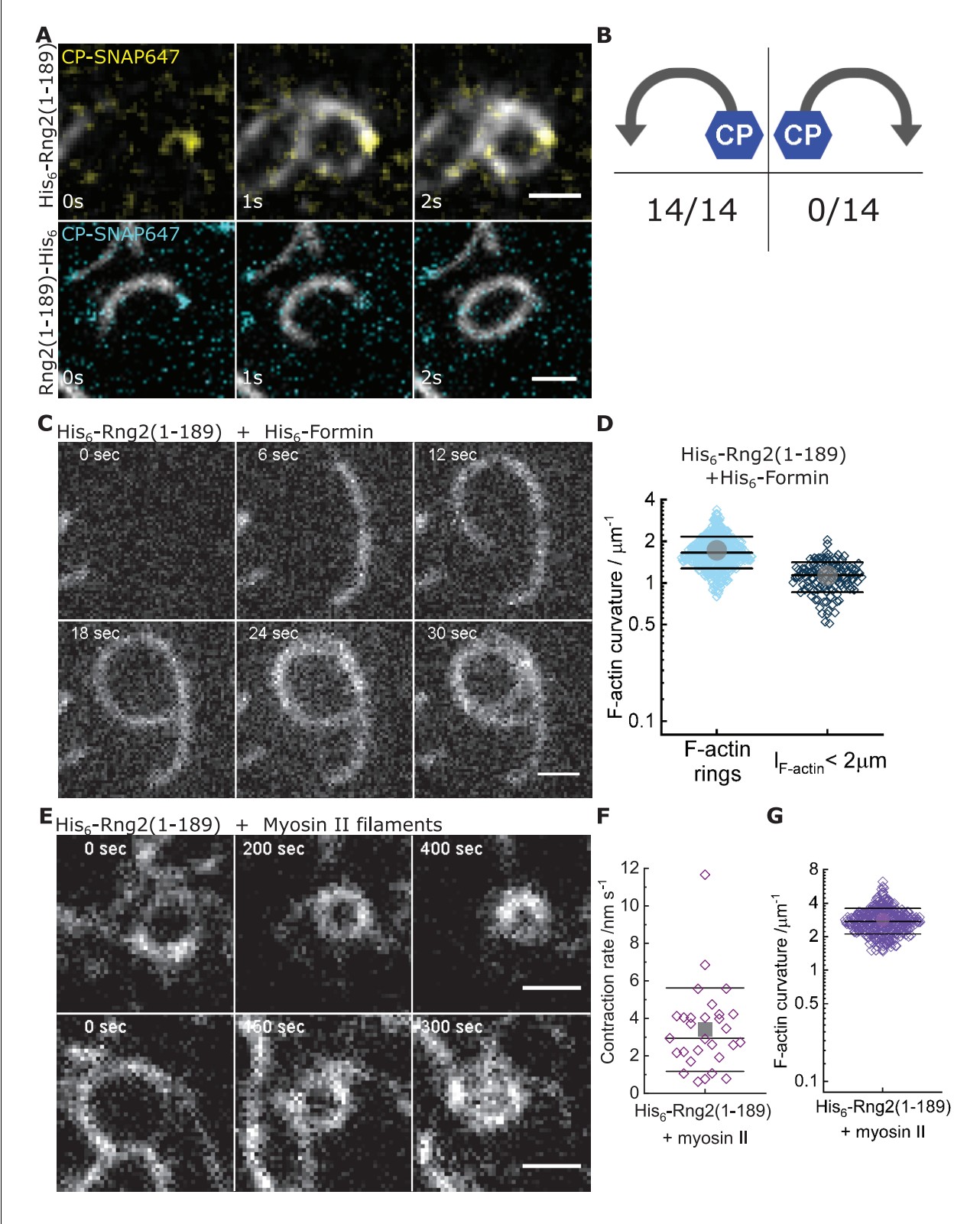

**Figure 3.** Curly recognizes actin filament orientation and enables actin ring contraction by myosin II. (**A**) TIRF microscopy images of actin filaments (Alexa488) ($C_{actin}$ = 100 nM) with the plus end marked with SNAP647-tagged capping protein ($C_{CP}$ = 2 nM) binding to $His_6$-curly (top) and curly-$His_6$ (bottom) ($C_{curly}$ = 10 nM); scale bar: 1 μm. (**B**) Count of actin filament bending orientations with respect to the capping protein where individual actin filaments could be identified. (**C**) TIRF microscopy images of a polymerizing actin filament (Alexa488) driven by membrane tethered $His_6$-formin in the

*Figure 3 continued on next page*

*Figure 3 continued*

presence of His$_6$-curly; scale bar: 1 µm. (**D**) Curvature measurements of actin filament rings (light blue) and curved short actin filaments (< 2 µm; gray-blue); shown are the individual data points and their mean ± s.d.; N$_{rings}$ = 477, N$_{short}$ = 125 from nine field of views of three independent experiments. (**E**) TIRF microscopy images of actin filament (Alexa488) ring contraction after addition of rabbit muscle myosin II filaments on His$_6$-curly containing SLBs; scale bar: 1 µm. (**F**) Average contraction rates of actin filament rings after addition of rabbit muscle myosin II filaments; shown are the individual data points and their mean ± s.d.; N = 18 from two individual experiments. (**G**) Curvature measurements of actin filament rings and curved segments 20 min after addition of rabbit muscle myosin II filaments; shown are the individual data points and their mean ± s.d.; N = 342 from 10 field of views of two individual experiments.

The online version of this article includes the following figure supplement(s) for figure 3:

**Figure supplement 1.** Curly recognizes actin filament orientation as visualised by labelled capping protein.
**Figure supplement 2.** Curly induces actin bending in membrane tethered formin-generated actin filaments.
**Figure supplement 3.** Tropomyosin supports actin bending by curly.
**Figure supplement 4.** Curly can bend actin filaments multiple times into a ring.
**Figure supplement 5.** Muscle myosin II filaments contract curly induced actin rings even tighter.

## Results and discussion

Our results show that the N-terminal CHD of IQGAP proteins induces actin filament bending when tethered to lipid membranes, which constitutes a new type of actin binding protein and could be an important link between actin and membrane geometries (for a summary of results, see *Supplementary file 1*). This mechanism of actin ring formation stands out as it bends single actin filaments in contrast to other reported systems that generate rings made of bundles of actin filaments (*Litschel et al., 2020*; *Mavrakis et al., 2014*; *Mishra et al., 2013*; *Miyazaki et al., 2015*; *Way et al., 1995*). The actin-binding affinity of Rng2(1-189) is Kd = 0.9 µM (*Hayakawa et al., 2020*) and CHDs of other proteins, such as α-actinin (K$_d$ = 4.7 µM; *Wachsstock et al., 1993*) or utrophin (Kd = 1.5 µM; *Singh et al., 2017*) do not show this behavior. Recently, Uyeda and colleagues reported that curly (Rng2(1-189)) in solution could induce kinks at random locations of the actin filament (*Hayakawa et al., 2020*). We do not know yet the precise mechanism how curly induces actin bending when constrained to a lipid membrane, but one possibility could be asymmetric binding of curly to actin filaments leading to a succession of kinks towards the same direction. With an estimated His$_6$-Curly surface density on SLBs of 5000 µm$^{-2}$ (*Nye and Groves, 2008*) the approximated curly to actin ratio would be 1:7 or higher. The mobility of curly on the SLB allowing accumulation under actin filaments is essential for actin filament bending as glass-immobilized curly failed to generate rings. Studies of the Goode laboratory on IQGAP1 indicate that the N-terminal part IQGAP1(1–522), which corresponds to Rng2(1-189), is monomeric and does not promote bundling of actin filaments (*Hoeprich et al., 2021*). The binding of curly to actin reduces the effective actin filament persistence length far below its typical value of 10 µm. This increased flexibility is particularly highlighted by the fact that addition of rabbit muscle myosin II filaments resulted in actin ring constriction without any evidence for filament breaking up to curvatures of 6.3 µm$^{-1}$ which is much higher than values reported for actin alone (C = 2.5 µm$^{-1}$) (*Taylor et al., 2000*). How this reduction of actin persistence length is achieved and whether it is related to the effect observed with cofilin (*De La Cruz and Gardel, 2015*; *McCullough et al., 2008*) remains to be deciphered in future work.

It was not evident that the addition of myosin II filaments would lead to actin ring constriction without having to add any actin cross-linkers. When considering that curly arranges actin filaments into an inward spiral, a possible explanation for actin ring constriction would be that the myosin II filament acts both as a cross-linker and motor protein: one end of the myosin II filament sits at the actin filament plus end while other myosin head domains of the same myosin II

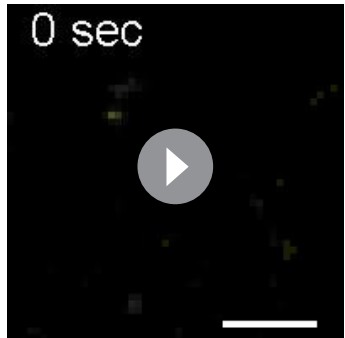

**Video 3.** Example image sequence of an actin filament (Alexa488, gray) with the plus end labeled by capping protein (SNAP647, yellow) landing on a His$_6$-curly decorated SLB; scale bar: 1 µm.
https://elifesciences.org/articles/61078#video3

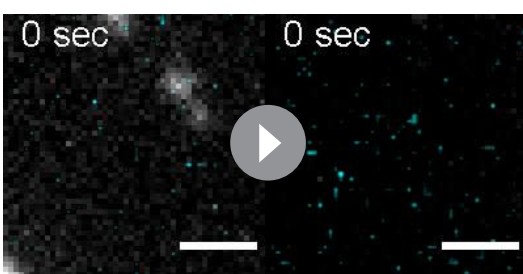

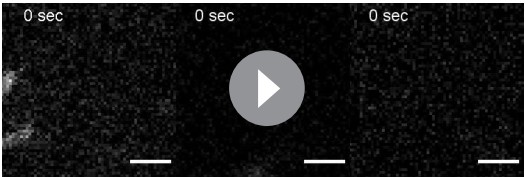

**Video 4.** Example image sequences of actin filaments (Alexa488, gray) with the plus end labeled by capping protein (SNAP647, cyan) landing on a curly-His$_6$ decorated SLB; scale bar: 1 μm.
https://elifesciences.org/articles/61078#video4

**Video 5.** Example image sequences of actin filaments (Alexa488) polymerized by SLB tethered formin in the presence of His$_6$-curly bound to the SLB; scale bar 1 μm.
https://elifesciences.org/articles/61078#video5

filament pull along the same actin filament to travel toward the plus end leading to constriction. This would result in sub-optimal myosin head orientations towards the actin filament, which could explain the observed slow constriction rates that were orders of magnitude slower than the reported values for actin propulsion by myosin II in motility assays (*Toyoshima et al., 1990*). Future works will provide more insights into this peculiar minimal ring constriction mechanism.

Highly bent actin filament structures are most likely important for many cellular structures such as axons (*Vassilopoulos et al., 2019*; *Xu et al., 2013*), mitochondrial actin cages (*Kruppa et al., 2018*), or secretion of the Weibel-Palade bodies (*Nightingale et al., 2011*), but the molecular mechanisms leading to their formation are still poorly understood. Future work on the role of curly in these processes could provide crucial insights into these mechanisms. In addition, our system of membrane-bound curly, actin filaments, and myosin II filaments constitutes a minimalistic system for actin ring formation and constriction and could be used in the future to design synthetic dividing vesicles and further exiting active membrane-cortex systems.

## Materials and methods

### Cloning and protein purification

*S. pombe* Rng2 fragments, Fim1, Cdc12 (FH1-FH2) and *S. cerevisiae* Iqg1 were amplified from cDNA library and genomic DNA, respectively. Amplified fragments were cloned into pET (His6) and pGEX (GST)-based vectors using Gibson cloning method (NEB builder, E5520S). The plasmids used in this study are listed in *Supplementary file 2*.

All protein expression plasmids were transformed into *E. coli* BL21-(DE3). Single colony was inoculated in 20 ml of LB media supplemented with appropriate antibiotic (pET-Kanamycin; pGEX-Ampicillin). Precultures were grown for ~12–16 hr at 36℃ shaking at 200 r.p.m. Cells were diluted to OD600 of 0.1 a.u. in 500 ml of LB with antibiotics and protein expression was induced with 0.25 mM isopropyl β-D-1-thiogalactopyranoside (IPTG). Protein was expressed for 3–4 hr at 30℃ shaking at 200 r.p.m. unless otherwise noted. After induction cell pellets were collected and spun down at 7000 r.p.m for 20 min after induction at 4℃. Media was aspirated and pellets were washed once with cold phosphate buffered saline (PBS) with 1 mM phenylmethylsulfonyl fluoride (PMSF), and pellets were stored at −80℃.

Ni-NTA purification of truncated IQGAP proteins (His6 tagged Rng2, Iqg1 and IQGAP1): Cell pellets for purification were thawed on ice for 10 min. The pellets were resuspended in 10 ml of lysis buffer for sonication (50 mM Napi pH 7.6,

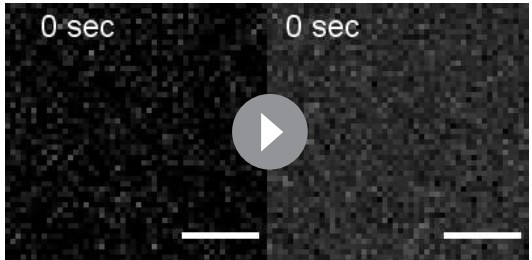

**Video 6.** Example image sequences of actin filaments (Alexa488) decorated with tropomyosin binding to membrane tethered His$_6$-curly; scale bar: 1 μm.
https://elifesciences.org/articles/61078#video6

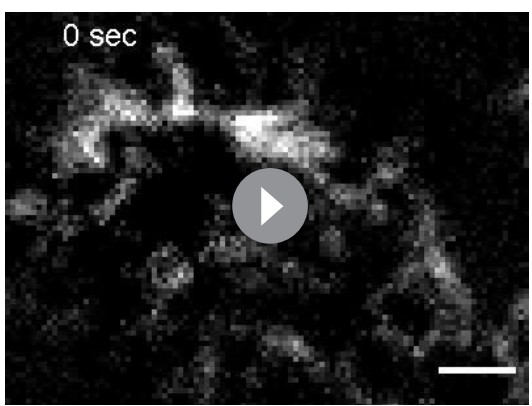

**Video 7.** Example image sequence showing formation, translation, and contraction of actin filament (Alexa488) rings on membrane tethered His$_6$-curly after the addition of muscle myosin II filaments; scale bar: 1 μm.

https://elifesciences.org/articles/61078#video7

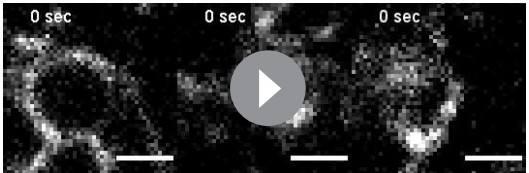

**Video 8.** Example image sequences of actin filament (Alexa488) ring contraction on membrane tethered His$_6$-curly after the addition of muscle myosin II filaments; scale bar: 1 μm.

https://elifesciences.org/articles/61078#video8

200 mM NaCl, 10 mM Imidazole pH 7.5, 0.5 mM EDTA, 1 mM DTT, 1 mg/ ml lysozyme, and complete mini-EDTA-Free protease inhibitor cocktail) and incubated on ice for 20 min, followed by sonication (eight cycles, 15 s pulse). The lysates were centrifuged at 14000 r.p.m, 30 min, 4°C and the clarified lysate was transferred to a 15 ml tube. The 400 μl slurry of HisPur Ni-NTA agarose resin (cat. no. 88221, Thermo fisher) was washed with wash buffer (5x) (50 mM Napi (pH 7.6), 300 mM NaCl, 30 mM Imidazole pH 7, 0.5 mM EDTA and 1 mM DTT) before the lysate was added. The clarified lysate was added to the washed Ni-NTA resin and incubated for 2 hr at 4°C. After incubation with NiNTA resin, beads were washed with wash buffer 6–8 times in poly-prep chromatography columns (cat. no. 7311550, BIO-RAD laboratories Inc). Protein was eluted using Ni-NTA elution buffer (50 mM NaPi pH 7.6, 300 mM NaCl, 0.5 mM EDTA, 1 mM DTT and 500 mM imidazole) and 300 μl elutions were collected in a clean Eppendorf tubes. Each fraction was assessed by SDS–polyacrylamide gel electrophoresis (SDS–PAGE). The eluates (E1-E3) were pooled, concentrated and buffer exchanged into the protein storage buffer (50 mM Tris-HCl pH 7.4, 150 mM NaCl, 1 mM DTT and 10% glycerol) using a PD MiniTrap G-25 sephadex columns (GE Healthcare, cat. No. GE28-9180-07) and the purified proteins were flash frozen in liquid N$_2$ and stored at −80°C. The protein concentration was estimated by UV280 and by comparing known quantities of BSA standards on an SDS–PAGE gel.

GST tagged protein (GST-Fim1) purification: Cell pellets for purification were thawed on ice for 10 min. The pellets were resuspended in 10 ml of lysis buffer for sonication (PBS, 0.5 mM EDTA, 1 mM DTT, 1 mg/ ml lysozyme, and complete mini-EDTA-Free protease inhibitor cocktail tablets) and incubated on ice for 20 min, followed by sonication (10 cycles, 15 s pulse). After sonication cell lysate was incubated with 1% Triton-X-100 for 20 min on ice. The lysates were centrifuged at 22,000xg, 30 min, 4°C and the clarified lysate was transferred to a 15 ml tube. The 400 μl slurry of glutathione sepharose-4B resin (cat. no. GE17-0756-01, GE) was washed with wash buffer (5x) (PBS, 0.5 mM EDTA and 1 mM DTT) before the lysate was added. The clarified lysate was added to the washed glutathione sepharose resin and incubated for 2–3 hr at 4°C. After incubation with sepharose resin, beads were washed with wash buffer 6–8 times in poly-prep chromatography columns. Protein was eluted using GST elution buffer (50 mM Tris-HCl pH8.0 and 10 mM glutathione). Purified protein sample was quantified and stored in the storage buffer as described above in the previous section.

Acetylation mimicking version of tropomyosin (ASCdc8) was expressed in BL21-DE3 and protein was purified by boiling and precipitation method as described earlier (*Palani et al., 2019*; *Skoumpla et al., 2007*). Purified tropomyosin was dialyzed against the storage buffer (50 mM NaCl, 10 mM imidazole, pH 7.5, and 1 mM DTT), flash frozen in liquid N2 and stored at −80°C.

SNAP labeling (SNAP-Surface 549, S9112S, NEB) of capping protein-beta and Rng2 1–189 was performed as per the manufactures protocol.

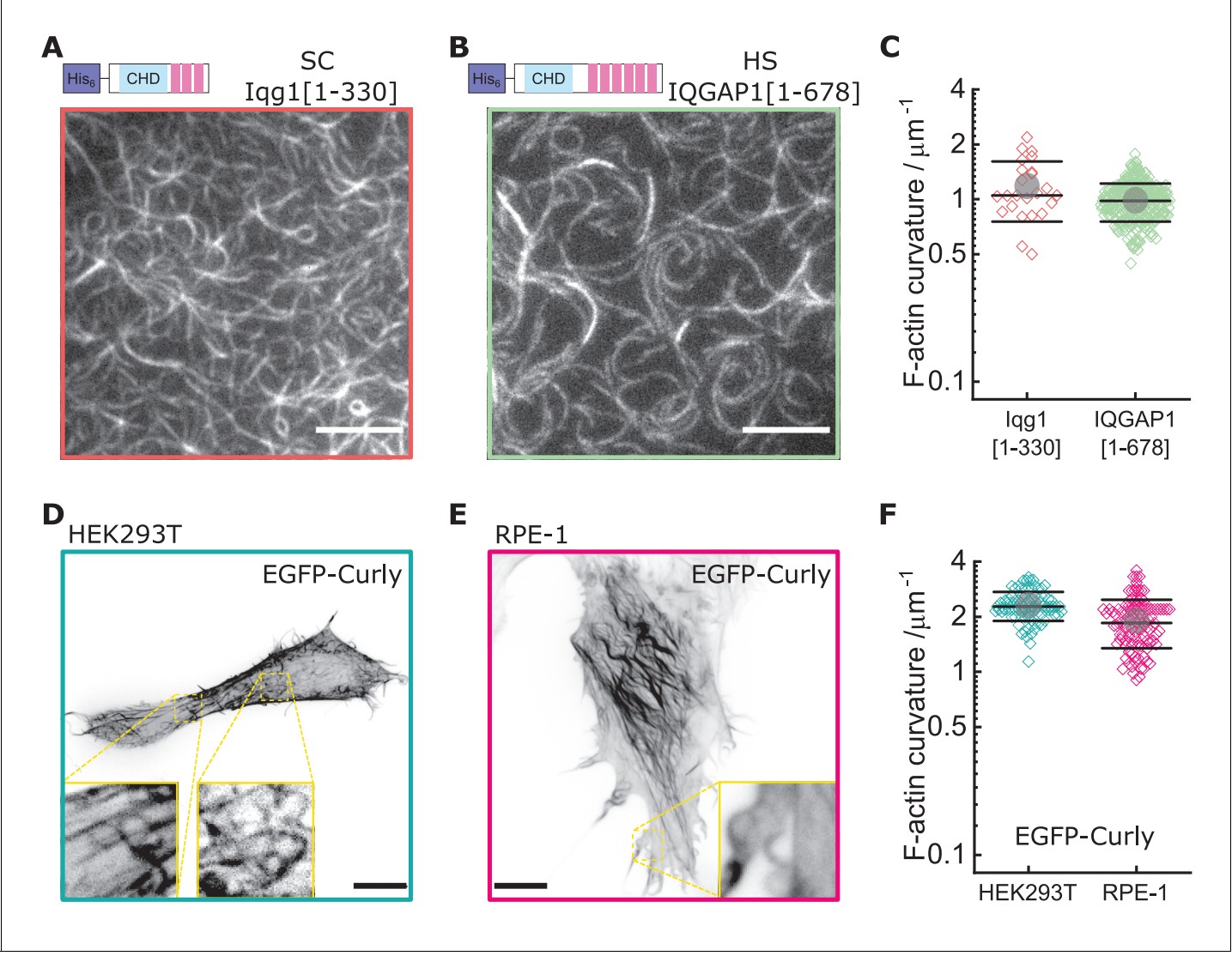

**Figure 4.** Curly effect is conserved among species and can foster actin bending in mammalian cells. (**A**) TIRF microscopy image of actin filaments (Alexa488) bound to membrane tethered His$_6$-Iqg1(1-330) (*S. cerevisiae*); image shows 1/9 of the field of view, scale bar: 5 µm. (**B**) TIRF microscopy image of actin filaments (Alexa488) bound to membrane tethered His$_6$-IQGAP1(1–678) (*H. sapiens*); image shows 1/9 of the field of view, scale bar: 5 µm. (**C**) Curvature measurements of actin filament rings and curved segments; shown are the individual data points and their mean ± s.d.; Iqg1(1-330) (orange): N = 110 from three field of views of each of three individual experiments; IQGAP1(1–678) (green): N = 407 from 20 field of views of three individual experiments. (**D**) Confocal microscopy image (maximum intensity projection of the basal cell section) of a HEK293T cell transfected with EGFP-Rng2(1-189), inlet shows zoom of dashed box; scale bar: 5 µm. (**E**) Confocal microscopy image (maximum intensity projection of the basal cell section) of a RPE-1 cell transfected with EGFP-Rng2(1-189), inlet shows zoom of dashed box; scale bar: 5 µm. (**F**) Curvature measurements of actin filament rings and curved segments found in EGFP-Rng2(1-189) expressing cells; shown are the individual data points and their mean ± s.d.; HEK293T (teal): N 91 from 14 cells of two independent experiments; REP-1 (fuchsia): N = 113 from 11 cells of two independent experiments.

The online version of this article includes the following figure supplement(s) for figure 4:

**Figure supplement 1.** The structures of S. Pombe Rng2(21-190) and H. Sapiens IQGAP1(28-290) are very similar.

**Figure supplement 2.** Curly induces actin rings in mammalian cells.

**Figure supplement 3, Curly induces highly curved actin filaments inside cells.** Lattice-light-sheet microscopy images (maximum intensity projections of middle or basal section) of HEK293T cells transfected with LifeAct-mCherry (magenta) and EGFP-Rng2(1-189) (cyan); green stars indicate location of ring structures; scale bar: 5 µm.

**Figure supplement 4.** Addition of a CAAX domain to address curly to the plasma membrane is contraproductive.

**Figure supplement 5.** Overexpression of full-length IQGAP1 also leads to the formation of highly curved actin filaments in cells.

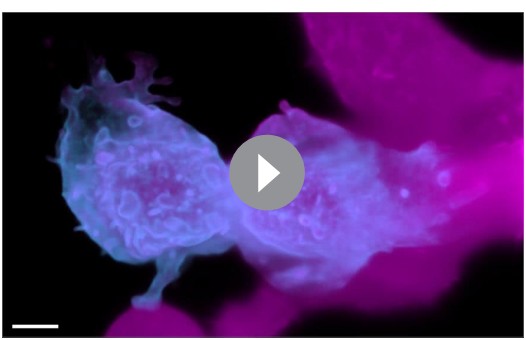

**Video 9.** 3D projection of HEK293T cell expressing EGFP-IQGAP1 (cyan) and Lifeact-mCherry (magenta); images obtained by lattice-light-sheet microscopy; scale bar: 10 µm.
https://elifesciences.org/articles/61078#video9

## SDS-PAGE

Purity of protein constructs was checked by running them on a 12% SDS-PAGE gel followed by staining with Coomassie blue (SimplyBlueStain, Invitrogen, cat. no. LC6060) and imaging on a ChemiDoc MP (BioRad).

## Mammalian cell expression and imaging

We used the mammalian cell lines HEK293T and RPE-1 obtained from ATCC which were tested negative for mycoplasma contamination. *S pombe* Rng2 fragments were cloned into pCDNA3.1-eGFP using the Gibson cloning method. HEK293T and RPE-1 cells were transiently transfected with pCDNA3 containing SpRng2(1-189), SpRng2(1-189)-CAAX, SpRng2 (144-189) or SpRng2(144-189)-CAAX using Lipofectamine 2000 (cat. no. 11668019, Life Technologies) following manufacturer's instructions. Cells were transfected at ~70% confluency for 24 hr before the experiments.

For confocal microscopy imaging, 500,000 cells were transfected with 0.5–1 µg of DNA, and eventually 0.5 µg of pTK93 Lifeact-mCherry (#46357, Addgene). Cells were seeded and imaged on µ-Dish 35 mm (cat. no. 81156, IBIDI). Before imaging, the culture medium was replaced with phenol red–free DMEM (Opti-MEM, cat. no. 31985062, Life Technologies). Images were taken using spinning disk microscope with a 100x Apo objective, NA 1.4 (Nikon Ti-E microscope equipped with Yokogawa spinning disk unit CSU-X and an Andor iXon 897 camera).

For lattice light sheet microscopy, 1 M cells were seeded in six well plates containing 5 mm cover glasses (#1.5 thickness) and transfected with either 0.1 µg pCDNA3.1-eGFP- SpRng2(1-189) or with 0.5 µg pEGFP-IQGAP1 (#30112, Addgene) and 0.5 µg pTK93 Lifeact-mCherry. Cells were imaged 16–22 hr post transfection. Cover glasses were mounted on the imaging chamber and DMEM medium was replaced by pre-warmed L-15 imaging medium (cat. no. 11415049, Gibco, Fisher scientific). Imaging was done at 37°C on a 3i second generation lattice-light-sheet microscope with a 0.71 NA LWD WI objective for excitation and a 1.1 NA WI objective for imaging providing a 62.5x magnification and equipped with 2 Hamamatsu ORCA-Flash 4.0v3 sCMOS cameras, resulting in a 104 nm pixel size. The sheet pattern was a Bessel lattice of 50 beams, with an inner and outer numerical aperture of 0.493 and 0.55 respectively. Sequential dual color imaging was performed using 488 nm and 561 nm lasers for excitation. 3D volumes were recorded with 0.57 µm step size (0.308 µm deskewed) for 150 planes with 100 ms exposure per plane.

## In vitro assay and total internal reflection fluorescence (TIRF) microscopy

### Supported lipid bilayer and experimental chamber preparation

The sample preparation, experimental conditions and lipid composition were similar to the ones described in previous work (*Köster et al., 2016*). Glass coverslips (#1.5 borosilicate, Menzel, cat. no. 11348503, Fisher Scientific) for SLB formation were cleaned with Hellmanex III (Hellma Analytics, cat. No. Z805939, Merck) following the manufacturer's instructions followed by thorough rinses with EtOH and MilliQ water and blow dried with N2 gas. For the experimental chamber, 0.2 ml PCR tubes (cat. no. I1402-8100, Starlab) were cut to remove the lid and conical bottom part. The remaining ring was stuck to the cleaned glass using UV glue (cat. no. NOA88, Norland Products) and 3 min curing by intense UV light at 265 nm (UV Stratalinker 2400, Stratagene). Freshly cleaned and assembled chambers were directly used for experiments.

Supported lipid bilayers (SLB) containing 98% DOPC (cat. no. 850375, Avanti Polar Lipids) and 2% DGS-NTA(Ni2+) (cat. no. 790404, Avanti Polar Lipids) lipids were formed by fusion of small unilamellar vesicles (SUV) that were prepared by lipid extrusion using a membrane with 100 nm pore

size (cat. no. 610000, Avanti Polar Lipids). SLBs were formed by addition of 10 µl of SUV mix (at 4 mM lipid concentration) to chambers filled with 90 µl KMEH (50 mM KCl, 2 mM MgCl$_2$, 1 mM EGTA, 20 mM HEPES, pH 7.2) and incubation for 30 min. Prior to addition of other proteins, the SLBs were washed 10 times by buffer exchange (always leaving 20 µl on top of the SLB to avoid damage by drying). We tested the formation of lipid bilayers and the mobility of lipids in control samples by following the recovery of fluorescence signal after photobleaching of hexa-histidine tagged GFP (His$_6$-GFP) as described in *Köster et al., 2016*.

## Actin filament polymerization and tethering to SLBs

Actin was purified from muscle acetone powder form rabbit (cat. no. M6890, Merck) and labeled with Alexa488-maleimide (cat. no. A10254, Thermo Fisher) following standard protocols (*Pardee and Spudich, 1982*).

In a typical experiment, actin filaments were polymerized in parallel to SLB formation to ensure that all components of the experiment were freshly assembled before starting imaging. First 10%$_{vol}$ of 10x ME buffer (100 mM MgCl$_2$, 20 mM EGTA, pH 7.2) were mixed with unlabeled and labeled G-actin (to a final label ratio of 20%), optionally supplemented with labeled capping protein in G-actin buffer (1 mM CaCl$_2$, 0.2 mM ATP, 2 mM Tris, 0.5 mM TCEP-HCl, pH 7.2) to a final G-actin concentration of 10 µM and incubated for 2 min to replace G-actin bound Ca$^{2+}$ ions with Mg$^{2+}$ ions. Polymerization of actin filaments was induced by addition of an equal amount of 2x KMEH buffer supplemented with 2 mM Mg-ATP bringing the G-actin concentration to 5 µM. After 30 min incubation time, actin filaments were added to the SLBs using blunt-cut pipette tips at a corresponding G-actin concentration of 100 nM (to ensure a homogenous mix of actin filaments, 2 µl of actin filament solution was mixed in 18 µl KMEH and then added to the SLB containing 80 µl KMEH). After 10 min of incubation, His$_6$-Curly or other variants of histidine-tagged actin binding proteins at a final concentration of 10 nM were added and a short time after (1–5 min) binding of actin to the SLB could be observed using TIRF microscopy.

In experiments with formin, the SLB was first incubated with 10 nM His$_6$-SpCdc12(FH1-FH2) and 10 nM His$_6$-Curly for 20 min, then washed twice with KMEH. During the incubation time, 10%$_{vol}$ of 10x ME buffer was mixed with unlabeled and labeled G-actin at 4 µM (final label ratio of 20%) together with 5 µM profilin and incubated for 5 min prior to addition to the SLB and imaging with TIRF microscopy.

In experiments with tropomyosin or fimbrin, actin filaments (C$_{G-actin}$ = 1 µM) were incubated with tropomyosin at a 1:3 protein concentration ratio or with fimbrin at a 3:2 protein concentration ratio for 15 min prior to addition to the SLB (*Palani et al., 2019*).

In experiments with rabbit muscle myosin II filaments, we prepared muscle myosin II filaments by diluting the stock of muscle myosin II proteins (rabbit, m. psoas, cat. no. 8326–01, Hypermol) (C$_{myoII}$ = 20 µM; 500 mM KCl, 1 mM EDTA, 1 mM DTT, 10 mM HEPES, pH 7.0) 10-times with MilliQ water to drop the KCl concentration to 50 mM and incubated for 5 min to ensure myosin filament formation. Myosin II filaments were further diluted in KMEH to 200 nM and added to the actin filaments bound to the SLB by His$_6$-Curly by replacing 1/10 of the sample buffer with the myosin II filament solution and supplemented with 0.1 mM Mg-ATP as well as a mix of 1 mM Trolox (cat. no. 648471, Merck), 2 mM protocatechuic acid (cat. no. 03930590, Merck), and 0.1 µM protocatechuate 3,4-dioxygenase (cat. no. P8279, Merck) to minimize photobleaching. To summarize, the final buffer composition was 50 mM KCl, 2 mM MgCl$_2$, 1 mM EGTA, 20 mM HEPES, 0.1 mM ATP, 1 mM Trolox, 2 mM protocatechuic acid and 0.1 µM protocatechuate 3,4-dioxygenase at pH 7.2 containing actin filaments (C$_{G-actin}$ = 100 nM) and myosin II filaments (C$_{myoII}$ = 20 nM). It was important to keep the pH at 7.2, as changes in pH would affect motor activity. As reported earlier, myosin filaments started to show actin network remodeling activity after about 10–15 min of incubation (*Köster et al., 2016*; *Mosby et al., 2020*).

## TIRF microscopy

Images were acquired using a Nikon Eclipse Ti-E/B microscope equipped with perfect focus system, a Ti-E TIRF illuminator (CW laser lines: 488 nm, 561 nm, and 640 nm) and a Zyla sCMOS 4.2 camera (Andor, Oxford Instruments, UK) controlled by Andor iQ3 software.

## Image analysis

Images were analyzed using ImageJ (http://imagej.nih.gov/ij).

Curvature was measured by fitting ellipses to match the actin filament contour by hand, while measuring first fully formed rings before curved actin filament segments and by going from the highest curvatures down to lower curvatures in each image with a cutoff for measurements at curvatures smaller than 0.1 $\mu m^{-1}$ or at 30–40 measurements per image (see examples in *Figure 1—figure supplement 1B*).

The actin ring contraction rate upon myosin II filament action was measured by generating kymographs based on a line (three pixels width) dividing the ring into two equal halves.

The 3D projection animation of HEK293 cells expressing EGFP-IQGAP1 and mRuby-LifeAct was generated with the 3Dscript plugin for ImageJ (*Schmid et al., 2019*).

## Data plotting and statistics

Graphs were generated using OriginPro (version 2019b, OriginLab, USA). All box plots depict individual data points, mean (circle), median (central line) and standard deviation (top and bottom lines).

## Acknowledgements

The authors thank Dr. Gayathri Panaghat (IISER Pune, India), Dr. Minhaj Sirajuddin (InStem, Bangalore, India), for insightful discussions and Prof. Gijsje Koenderink (TU Delft, Netherlands), Prof. Rob Cross (University of Warwick, UK) for helpful comments on the manuscript, and Prof. Satyajit Mayor and Prof. Laurent Blanchoin for kindly sharing the Ezrin-ABD and α-actinin protein expression constructs, respectively. The work was supported by a Wellcome Investigator Award (WT 101885MA) and an ERC advanced grant (ERC-2014-ADG N° 671083) to MKB. DVK thanks the Wellcome-Warwick Quantitative Biomedicine Programme for funding (Wellcome Trust ISSF, RMRCB0058). SP thanks the DBT-IISc partnership funds and IISc start-up grant. SG was supported by an international Chancellor's fellowship of the University of Warwick. SC was supported by a research development grant of the University of Warwick. EIM was supported by the A*STAR Research Attachment Programme (ARAP) PhD studentship. The authors thank Helena Coker of CAMDU (Computing and Advanced Microscopy Unit) and for her support & assistance in this work. The Warwick Lattice Light Sheet microscope is funded by Wellcome (208384/Z/17/Z).

## Additional information

### Competing interests

Mohan K Balasubramanian: Reviewing editor, *eLife*. The other authors declare that no competing interests exist.

### Funding

| Funder | Grant reference number | Author |
| --- | --- | --- |
| Wellcome Trust | WT 101885MA | Mohan K Balasubramanian |
| H2020 European Research Council | ERC-2014-ADG N° 671083 | Mohan K Balasubramanian |
| Wellcome Trust | ISSF-Warwick QBP RMRCB0058 | Darius Vasco Köster |
| Department of Biotechnology, Ministry of Science and Technology, India | DBT-IISc partnership grant | Saravanan Palani |
| University of Warwick | Research development fund - RD19012 | Scott Clarke |
| University of Warwick | International Chancellor's Fellowship | Sayantika Ghosh |
| University of Warwick | ARAP fellowship | Esther Ivorra-Molla |

The funders had no role in study design, data collection and interpretation, or the decision to submit the work for publication.

## Author contributions

Saravanan Palani, Conceptualization, Resources, Data curation, Formal analysis, Funding acquisition, Investigation, Methodology, Writing - original draft, Project administration, Writing - review and editing; Sayantika Ghosh, Esther Ivorra-Molla, Investigation, Mammalian cell culture and transfections; Scott Clarke, Investigation, Lattice-light-sheet microscopy; Andrejus Suchenko, Investigation, Help in protein purification; Mohan K Balasubramanian, Conceptualization, Supervision, Funding acquisition, Validation, Methodology, Writing - original draft, Project administration; Darius Vasco Köster, Conceptualization, Resources, Data curation, Formal analysis, Supervision, Funding acquisition, Validation, Investigation, Visualization, Methodology, Writing - original draft, Project administration, Writing - review and editing

## Author ORCIDs

Saravanan Palani (iD) http://orcid.org/0000-0002-1893-6777
Sayantika Ghosh (iD) https://orcid.org/0000-0003-1399-1168
Mohan K Balasubramanian (iD) https://orcid.org/0000-0002-1292-8602
Darius Vasco Köster (iD) https://orcid.org/0000-0001-8530-5476

## Decision letter and Author response

Decision letter https://doi.org/10.7554/eLife.61078.sa1
Author response https://doi.org/10.7554/eLife.61078.sa2

# Additional files

## Supplementary files

• Supplementary file 1. Table summarizing the experimental results.

• Supplementary file 2. List of the plasmids used in this study.

• Transparent reporting form

## Data availability

All data generated or analysed during the study are included in the manuscript and supporting files. Source data files for the actin curvature measurements (Figures 1-4) have been deposited and are freely available on Dryad (http://doi.org/10.5061/dryad.bvq83bk6f).

The following dataset was generated:

| Author(s) | Year | Dataset title | Dataset URL | Database and Identifier |
|---|---|---|---|---|
| Köster DV | 2020 | Calponin-Homology Domain mediated bending of membrane associated actin filaments | http://doi.org/10.5061/dryad.bvq83bk6f | Dryad Digital Repository, 10.5061/dryad.bvq83bk6f |

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

# Appendix 1

**Appendix 1—key resources table**

| Reagent type (species) or resource | Designation | Source or reference | Identifiers | Additional information |
|---|---|---|---|---|
| Gene (*Schizosaccharomyces pombe*) | Rng2 | Pombase | SPAC4F8.13c | |
| Gene (*Saccharomyces cerevisiae*) | Iqg1 | Saccharomyces Genome Database (SGD) | SGD: S000006163 | |
| Strain, strain background (*Escherichia coli*) | BL21(DE3) | New England Biolabs | C2527I | Chemical competent cells |
| Cell line (*Homo sapiens*) | Human Embryonic Kidney (HEK) 293T | ATCC | RRID:CVCL_0063 | |
| Cell line (*Homo sapiens*) | Retinal Pigment Ephitilial-1 (hTERT-RPE1) | ATCC | RRID:CVCL_4388 | |
| Recombinant DNA reagent | pTK93 Lifeact-mCherry | Unpublished (Dr Iain Cheeseman's lab) | RRID: Addgene46357 | mCherry version of LifeAct peptide |
| Recombinant DNA reagent | pEGFP-IQGAP1 | Ren et al J Biol Chem. 2005 Oct 14. 280 (41):34548–57. | RRID: Addgene30112 | EGFP version of Hs IQGAP1 |
| Recombinant DNA reagent | pET28C (+) | Novagen-Sigma Aldrich | Cat. #: 69866 | |
| Recombinant DNA reagent | pGEX-4T1 | Cytiva (GE Healthcare) | Cat. #: 28-9545-49 | |
| Recombinant DNA reagent | pET-3d-6HIS-SNAP-tagged β1 subunit and untagged α1 subunits of chicken CapZ | *Bombardier et al., 2015* | RRID: Addgene69948 | SNAP tagged version of Capping proteins |
| Recombinant DNA reagent | pGEX-alpha actinin4 (acnt4) | *Bombardier et al., 2015* *Ennomani et al., 2016* (Kind gift from Blanchoin's lab) | | 6HIS tagged version of Hsalpha-actinin4 |
| Recombinant DNA reagent | pET28C-6HIS-Rng2(1-189) | This paper | | 6HIS tagged version of SpRng2 (1-189) (*Supplementary file 2*) |
| Recombinant DNA reagent | pET28C-6HIS-Rng2(1-250) | This paper | | 6HIS tagged version of SpRng2 (1-250) (*Supplementary file 2*) |
| Recombinant DNA reagent | pET28C-6HIS-Rng2(1-300) | This paper | | 6HIS tagged version of SpRng2 (1-300) (*Supplementary file 2*) |
| Recombinant DNA reagent | pETMCN-Rng2 (1-189)-C-6HIS | This paper | | Cterminal −6HIS tagged version of SpRng2 (1-189) (*Supplementary file 2*) |

*Continued on next page*

*Appendix 1—key resources table continued*

| Reagent type (species) or resource | Designation | Source or reference | Identifiers | Additional information |
|---|---|---|---|---|
| Recombinant DNA reagent | pET23a-10HIS-SNAP-Rng2(1-300) | This paper | | 10HIS-SNAP tagged version of SpRng2 (1-300) (*Supplementary file 2*) |
| Recombinant DNA reagent | pET28C-6HIS-Sclqg1 (1-330) | This paper | | 6HIS tagged version of Sclqg1 (1-330) (*Supplementary file 2*) |
| Recombinant DNA reagent | pET28C-6HIS-HsIQ GAP1 (1–678) | This paper | | 6HIS tagged version of HsIQGAP1 (1–678) (*Supplementary file 2*) |
| Recombinant DNA reagent | pET28C-6HIS-Cdc12 (740–1391) | This paper | | 6HIS tagged version of SpCdc121 (740–1391) (*Supplementary file 2*) |
| Recombinant DNA reagent | pETMCN-AScdc8 | *Palani et al., 2019*, Journal of Cell Biology | | untagged version of acetyl mimicking SpCdc8 (*Supplementary file 2*) |
| Recombinant DNA reagent | pGEX4T1-GST-Fim1 | This paper | | GST tagged version of SpFim1 (*Supplementary file 2*) |
| Recombinant DNA reagent | pET23a-10HIS-SNAP-Ezrin-ABD | Unpublished, (Satyajit Mayor's lab) | | 10HIS-SNAP tagged version of Ezrin-ABD (*Supplementary file 2*) |
| Recombinant DNA reagent | pCDNA3-EGFP-GSGG-Rng2(1-189) | This paper | | EGFP tagged version of SpRng2 (1-189) (*Supplementary file 2*) |
| Commercial assay or kit | Gibson cloning (NEBuilder) | New England Biolabs (NEB) | Cat. #: E5520S | |
| Commercial assay or kit | HisPur Ni-NTA agarose resin | ThermoFisher | Cat. #: 88221 | |
| Commercial assay or kit | Glutathione Sepharose 4B | Cytiva (GE healthcare) | Cat. #: 17-0756-01 | |
| Commercial assay or kit | PD MiniTrap G-25 | Cytiva (GE healthcare) | Cat. #: 28918007 | |
| Chemical compound, drug | SNAP-Surface 549 | New England Biolabs (NEB) | Cat. #: S9112S | |
| Chemical compound, drug | SimplyBlue safe Stain | Invitrogen | Cat. #: LC6060 | |
| Other | Opti-MEM | Life Technologies | Cat. #: 31985062 | Reduced Serum Cell medium |
| Chemical compound, drug | Alexa488-maleimide | ThermoFisher | Cat. #: A10254 | |
| Other | poly-prep chromatography columns | Bio-Rad | Cat. #: 7311550 | Column for protein purification |
| Peptide, recombinant protein | Myosin II (Rabbit m. psoas) | Hypermol | Cat. #: 8326–01 | |
| Other | muscle acetone powder form rabbit | Merck (Sigma-Aldrich) | Cat. #: M6890 | Acetone powder from rabbit muscle used as a source for actin purification |
| Chemical compound, drug | 1,2-dioleoyl-sn-glycero-3-phosphocholine (DOPC) | Avanti Polar Lipids, RRID:SCR_016391 | Ca.t #: 850375 | |

*Continued on next page*

*Appendix 1—key resources table continued*

| Reagent type (species) or resource | Designation | Source or reference | Identifiers | Additional information |
|---|---|---|---|---|
| Chemical compound, drug | 1,2-dioleoyl-sn-glycero-3-[(N-(5-amino-1-carboxypentyl) iminodiacetic acid) succinyl] (nickel salt) (DGS-NTA(Ni)) | Avanti Polar Lipids, RRID:SCR_016391 | Cat. #: 790404 | |
| Other | Lipid extruder | Avanti Polar Lipids, RRID:SCR_016391 | Cat. #: 610000 | Tool used for the formation of small unilamellar vesicles (SUVs). |
| Other | µ-Dish 35 mm | IBIDI | Cat. #: 81156 | Cell culture dishes with glass bottom made for fluorescence microscopy. |
| Other | 24 mm x50 mm glass coverslips, #1.5, borosilicate | Menzel/ Fisher Scientific | Cat. #: 11348503 | Basis for the formation of lipid bilayers imaged via fluorescence microscopy. |
| Software, algorithm | ImageJ | NIH | RRID:SCR_003070 | Version 1.53 c |
| Primers | | | | |
| Sequence-based reagent | SpRng2_1Fw | This paper | PCR primers | CGGGATCCCGATGGACGTAA ATGTGGGATTATC |
| Sequence-based reagent | SpRng2_189Rv | This paper | PCR primers | CCGCTCGAGTCATTAAGCTTTGA AGTTAGGAAGGATTAC |
| Sequence-based reagent | SpRng2_250Rv | This paper | PCR primers | CCGCTCGAGTTAGTCTGAACGA GCGCTAGCATC |
| Sequence-based reagent | SpRng2_300Rv | This paper | PCR primers | CCGCTCGAGTCATTAAGATCGTT GCATATGTCCC |
| Sequence-based reagent | Sclqg1-1Fw | This paper | PCR primers | CGGGATCCCGATGACTGCCTA CTCCGGTTCC |
| Sequence-based reagent | Sclqg1-330Fw | This paper | PCR primers | CCGCTCGAGTTACACGTCAAGA TTGCTCATTTTAGG |
| Sequence-based reagent | SpFim1-Fw | This paper | PCR primers | CGGGATCCCGGATGTTAGCTCT TAAACTTCAAAAG |
| Sequence-based reagent | SpFim1-Rv | This paper | PCR primers | CCGCTCGAGTTATACGGCCA TTAAACTGCC |
| Sequence-based reagent | SpCdc12-740Fw | This paper | PCR primers | CGGGATCCCGATGGGCTCAACTA ATTCCAAGGAAAGG |
| Sequence-based reagent | SpCdc12-1391Rv | This paper | PCR primers | CCGCTCGAGTTAATTGTTGACA AGATTCAAACGTC |

