## [Decision Letter]

**Acceptance summary:**

This manuscript reports the structural effects of a fragment of the IQGAP family proteins, called "Curly", on actin filaments. When tethered to a supported lipid bilayer, Curly induces curvature in actin filaments, ultimately giving rise to ring-shaped filament structures. Moreover, this study demonstrates that filament decoration by tropomyosin increases the propensity of ring formation, and introduction of myosin II filaments induces constriction of actin rings. The findings presented in the manuscript are important in the cytoskeleton, cell division and cell morphogenesis fields, and the authors have addressed the points raised by reviewers in the revised manuscript.

**Decision letter after peer review:**

Thank you for submitting your article "Calponin-Homology Domain mediated bending of membrane associated actin filaments" for consideration by *eLife*. Your article has been reviewed by 3 peer reviewers, including Pekka Lappalainen as Reviewing Editor and Reviewer #1, and the evaluation has been overseen by Vivek Malhotra as the Senior Editor.

The reviewers have discussed the reviews with one another and the Reviewing Editor has drafted this decision to help you prepare a revised submission.

As the editors have judged that your manuscript is of interest, but as described below additional data are required in order to move forward with the publication. We would like to draw your attention to changes in our revision policy that we have made in response to COVID-19 (https://elifesciences.org/articles/57162). First, because many researchers have temporarily lost access to the labs, we will give authors as much time as they need to submit revised manuscripts. We are also offering, if you choose, to post the manuscript to bioRxiv (if it is not already there) along with this decision letter and a formal designation that the manuscript is "in revision at eLife". Please let us know if you would like to pursue this option. (If your work is more suitable for medRxiv, you will need to post the preprint yourself, as the mechanisms for us to do so are still in development.)

Summary:

This manuscript reports the structural effects of a fragment of the IQGAP family proteins, called "curly", on actin filaments. When tethered to a supported lipid bilayer, Curly induces curvature in actin filaments, ultimately giving rise to ring-shaped filament structures. Moreover, this study demonstrates that filament decoration by tropomyosin increases the propensity of ring formation, and introduction of myosin II filaments induces constriction of actin rings.

The findings presented in this manuscript are potentially very important, Howver, in some cases the results are somewhat preliminary and lack essential controls. Thus, additional experiments and data analysis are required to strengthen the study.

Essential revisions:

1. More thorough analysis of actin-interactions of Curly constructs should be done. The authors discuss (pages 1 -2) interactions of full-length Curly and its various deletion constructs with actin filaments. However, actin-binding was not properly tested for any of the constructs used in this study. Thus, the authors should carry out better actin filament co-sedimentation assays with all constructs. The assays should be performed with a constant concentration of Curly, and varying the actin concentration (form 0 μm to e.g. 8 uM) to obtain binding curves, and to be able to compare the affinities of different constructs for actin filaments. Given that Curly is proposed to contain two actin-binding sites, the authors should also test, or at minimum discuss, if this protein bundles filaments. Also, do multiple filaments ever become incorporated into the same ring? Finally, the authors should examine if various curly constructs used in the study are monomeric of dimeric, because this may shed light on the mechanism by which Curly induces curvature on actin filaments.

2. The cell biology data presented in Figure 4 and Figure 4 —figure supplement 2 are not particularly convincing. The authors should perform a careful quantification of F-actin curvature and 'actin ring frequency' in cells transfected with plasmids expressing (i). EGFP, (ii). EGFP-Curly, and (iii). an EGFP-Curly mutant defective in in ring formation. Additionally, are the cellular rings a similar size to those observed in vitro? Because EGFP-Curly most likely does not associate with the plasma membrane in cells, it is somewhat confusing how it can still induce the formation of actin rings. The authors may observe much more robust actin ring formation in cells if they would use a membrane-anchored Curly-EGFP instead of soluble EGFP-Curly. This should be, at minimum, discussed in the manuscript.

3. The visual components of this work are striking. However, the accompanying quantification is somewhat confusing. Throughout the text, mean values are listed for various parameters beyond those shown in the figures. It would improve the flow of the manuscript/aid the reader, if these were represented as panels in each figure. Further, at least 3 FOVs should be analyzed in all cases, from independent experiments. However, it appears that a single FOV was measured in several figures (i.e. Figure 3 sup 1; Figure 3 sup 2). Other experiments also have relatively low "n" (i.e. 6 filaments measured for the analysis in Figure 2 sup 2). The authors should confirm that the "n" values have enough statistical power to support the conclusions.

[Editors' note: further revisions were suggested prior to acceptance, as described below.]

Thank you for submitting your article "Calponin-Homology Domain mediated bending of membrane associated actin filaments" for consideration by *eLife*. Your article has been reviewed by 3 peer reviewers, including Pekka Lappalainen as Reviewing Editor and Reviewer #1, and the evaluation has been overseen by Vivek Malhotra as the Senior Editor.

Essential Revisions:

The actin-binding experiments are not particularly convincing, and the current data do not provide strong support for authors' conclusions about two actin-binding sites in curly. Therefore, the authors should delete all co-sedimentation data and comments about different actin binding sites from the manuscript.

*Reviewer #1:*

This is a revised version of a manuscript providing evidence that, when anchored to the membrane, 'curly' region of IQGAP proteins can stabilize actin filaments in a curved geometry. The manuscript is improved, and the authors have addressed many of my previous concerns. However, there are still few issues that should be addressed to strengthen the manuscript.

1. Based on the data presented in Figure 1 – supplement 1A, the quality of many protein preps is not sufficient for reliable biochemical experiments. The His6-Rng2 fragments contain huge amount of impurities, and are thus not suitable for biochemcial experiments. Thus, it is strongly recommended that the authors would either delete all data obtained by using these proteins from the manuscript, or alternatively make much better quality protein preps and repeat the experiments with these.

2. The actin filament co-sedimentation assay presented in Figure 1 – supplement 1B and C is not very convincing. I do not understand why the authors carried out this assay with a constant actin concentration and varying the concentration of Rng2(1-189). This is because much more reliable way to carry out such assay would be to keep the concentration of Rng2(1-189) constant, and vary the concentration of actin. This way the possible spontaneous aggregation of Rng2(1-189) would not affect the interpretation of the results. Therefore, these inconclusive data should be deleted from the manuscript. Unless the authors can measure the affinities of various protein constructs for F-actin, they should remove all speculations about different actin-binding sites from the manuscript text (lines 51-54, 63-84, and 195-198).

3. I am still not fully convinced about the CP-SNAP647 experiments (presented in Figure 3A and Figure 3 – supplement 1). This is because, in addition to CP-dots at filament ends, also other CP-SNAP647 dots not associated with the filament ends are visible in the images. Moreover, the supplementary video examples are not particularly convincing/high quality. Thus, the authors may also consider removing these data from the manuscript.

*Reviewer #2:*

In this manuscript, Palani and coworkers investigate the structural effects of binding of a fragment of the IQGAP family of proteins, called "curly", to actin filaments. When tethered to a supported lipid bilayer, curly induces curvature in actin filaments, ultimately giving rise to ring-shaped filament structures. Filament decoration by tropomyosin increases the propensity of ring formation, and introduction of myosin II filaments induces constriction.

This manuscript presents novel and intriguing insights into the mechanisms that regulate the formation of cytoskeletal structures with curved geometries. The manuscript is well written, and the experiments are logically described. As such, this paper is sure to be of interest to a broad audience.

The authors have addressed all of the comments I made in the first round of reviews. I have a few questions remaining, as detailed below:

(1) I am confused about the y-axis scale on the graph of the amount of filamentous actin each Rng2 construct tethered to the bilayers that the authors included in their response to "Essential Revisions" Question 1. What does it mean for the value of Aactin/Atotal to be 100? Is this graph different from the graph included in Figure 2 Supplement 1 Part B?

(2) I find it curious that the authors did not perform a low-speed co-sedimentation assay with their curly construct to test for filament bundling activity. This would have provided a more quantitative basis for their discussion of whether multiple actin filaments might ever be incorporated into the same ring structure.

(3) In response to my question about a schematic of a myosin II-bound actin ring (originally in Figure 3 Supplement 4 Part E), the authors state that they have made modifications to the model. However, I am unable to locate this schematic in the revised manuscript. Did the authors remove it?

(4) In response to my question about the effects of the counterclockwise curvature on the structure of the actin filaments, the authors state: "However, data by XXX indicates that binding of curly can loosen/disrupt the helical structure of actin filaments…". Who is XXX?

(5) In response to my question about binding/structural cooperativity, the authors state that they do not have direct evidence to determine whether binding of curly to actin filaments is cooperative, or if filament curvature arises from an additive effect of multiple independent binding events. To reflect this fact, they indicate that they have rephrased the following (original) statement using more cautious language:

"Importantly, the uni-directional bending supports the hypothesis that the binding site of curly with actin filaments defines an orientation, and the propagation of a curved trajectory once established indicates a cooperative process."

However, their revised statement still appears to suggest a cooperative model for binding/bending without acknowledging the possibility of other binding models.

(6) The authors mention that they have added a quantification of ring frequency in curly-expressing cells compared to cells expressing only Lifeact-mCherry. Perhaps I have misunderstood their graphs, but I was unable to locate this analysis (or any micrographs or analysis with either wild-type, "EGFP alone" or "Lifeact-mCherry alone" cells).

*Reviewer #3:*

The authors have addressed all my previous concerns (reviewer 3 in the rebuttal).

However, I think the authors have left a place holder in their response to reviewer 2 (However, data by XXX indicates that binding of curly can loosen/ disrupt the helical structure of actin filaments leading to kinks. This would indicate towards the 2nd suggested mechanisms), as I do not know who XXX is meant to indicate.

---

## [Author Response]

Essential revisions:1. More thorough analysis of actin-interactions of Curly constructs should be done. The authors discuss (pages 1 -2) interactions of full-length Curly and its various deletion constructs with actin filaments. However, actin-binding was not properly tested for any of the constructs used in this study. Thus, the authors should carry out better actin filament co-sedimentation assays with all constructs. The assays should be performed with a constant concentration of Curly, and varying the actin concentration (form 0 μm to e.g. 8 uM) to obtain binding curves, and to be able to compare the affinities of different constructs for actin filaments. Given that Curly is proposed to contain two actin-binding sites, the authors should also test, or at minimum discuss, if this protein bundles filaments. Also, do multiple filaments ever become incorporated into the same ring? Finally, the authors should examine if various curly constructs used in the study are monomeric of dimeric, because this may shed light on the mechanism by which Curly induces curvature on actin filaments.

We have performed sedimentation assay with Rng2(1-189) and with Rng2(150-250) as these two constructs are the most important ones for the study. Full length, because it bends actin into rings and the Rng2(150-250) since it contains a previously unreported actin binding region. A problem was that inspection of Coomassie-stained gels of our truncations of Rng2(1-189) displayed low expression and multiple bands indicating protein instability and degradation. This made it impractical for us to quantify their actin binding affinity using the co-sedimentation assay (added to Figure 2-sup Figure 1).

However, we could find a binding constant of Kd = 1 μM for Rng2(1-189) similar to the value reported by Hayakawa et al., (2020). Unfortunately, the Rng2(150-250) construct aggregated and accumulated in the pellet which made it impossible to reliably estimate a binding coefficient for the construct (added to Figure 1-sup Figure 1).

As an alternative to the co-sedimentation assay, we included a quantification of the amount of F-actin that each construct tethered to SLBs to compare binding affinities of the different constructs. (added to Figure 2-sup Figure 2)

We do not have evidence that rng2(1-189) is a potent actin bundling protein when tethered to SLBs as most tethered actin filaments were single and did not bundle. However, any of the apparent actin bundles were straight and did not display high curvatures. Of all constructs studied here the construct with the highest propensity to induce actin bundling was *S. cerevisiae* -iqg1[1-330].

Regarding whether curly is a monomer or dimer, we revisited our data using fluorescently labelled curly, and did not find evidence for dimer formation as the majority of the labelled protein covered the SLB homogeneously and no bright puncta were observed. However, we did not have the means to perform fluorescence correlation spectroscopy to quantify the number of monomers vs. multimers on the SLB and would intend to do this in future studies.

It is a good question whether multiple actin filaments incorporate into the same ring structure. We observed that the same actin filaments can wind around the same ring multiple times. Given the low actin densities used in the study, it is very rare that an actin filament landed on top of a ring, but we would not rule out that multiple actin filaments form along the same ring. However, given limitations of fluorescence microscopy, we would not be able to say whether this is due to the binding of one actin filament to another one promoted by rng2(1-189). Future studies using e.g. EM would be very helpful in answering this question.

We added these aspects in the discussion.

2. The cell biology data presented in Figure 4 and Figure 4 —figure supplement 2 are not particularly convincing. The authors should perform a careful quantification of F-actin curvature and 'actin ring frequency' in cells transfected with plasmids expressing (i). EGFP, (ii). EGFP-Curly, and (iii). an EGFP-Curly mutant defective in in ring formation. Additionally, are the cellular rings a similar size to those observed in vitro? Because EGFP-Curly most likely does not associate with the plasma membrane in cells, it is somewhat confusing how it can still induce the formation of actin rings. The authors may observe much more robust actin ring formation in cells if they would use a membrane-anchored Curly-EGFP instead of soluble EGFP-Curly. This should be, at minimum, discussed in the manuscript.

We did only show a few cell images as we only wanted to show that Curly affects the cell cortex, and we agree that more data would be helpful. We have now imaged cells expressing lifeact-mCherry as an actin marker together with EGFP-Curly, EGFP alone, EGFP-rng2(150-250) as well as membrane targeted protein constructs using the CAAX-motif using lattice light sheet microscopy. As reported in the previous version of the manuscript, EGFP-Curly expression leads to a higher occurrence of highly bent actin filaments. A possible explanation could be that the additional curly enhances the effect of endogenous IQGAP1 in the cells and binds preferentially at highly curved actin filaments (Figure 4, Figure 4.-supplement Figure 2).

The EGFP-rng2(150-250) construct was mainly cytoplasmic. We also thought that membrane anchored curly would induce stronger actin ring formation in cells. However, adding the CAAX domain did result in high cell death and very little protein location at the plasma membrane. Interestingly, colleagues in the Mayor laboratory at NCBS have also observed similar problems when expressing e.g. the ezrin actin binding domain linked to a transmembrane protein pointing to a general problem of membrane tethered proteins with high affinity to actin. In future, we are planning to use e.g. the rapamycin inducible FRB-FKBP linker system to target curly to the plasma membrane in an efficient and controlled way, but this would go beyond the scope of this paper (this is now part of the discussion).

We quantified the actin curvatures in cells expressing EGFP-curly (Figure 4F in the previous manuscript) that turn out to be very similar to the ones found in vitro, we have added a quantification of ring frequency in curly expressing cells compared to cells expressing lifeact-mCherry only expressing ones.

3. The visual components of this work are striking. However, the accompanying quantification is somewhat confusing. Throughout the text, mean values are listed for various parameters beyond those shown in the figures. It would improve the flow of the manuscript/aid the reader, if these were represented as panels in each figure. Further, at least 3 FOVs should be analyzed in all cases, from independent experiments. However, it appears that a single FOV was measured in several figures (i.e. Figure 3 sup 1; Figure 3 sup 2). Other experiments also have relatively low "n" (i.e. 6 filaments measured for the analysis in Figure 2 sup 2). The authors should confirm that the "n" values have enough statistical power to support the conclusions.

We thought to highlight the data on filament curvature because of which we plotted those as graphs. As suggested, we have now added accompanying panels in the figures for the other quantitations. To ease the reading flow, we have removed the stated values of the key parameters (curvature, number of rings, number of samples) from the main text and added a table summarising them. In addition, the relevant numbers are also listed in the figure legends.

Regarding the statistics, we are very sorry that the reviewers were of the impression that only a single FOV was analysed. We have analysed a minimum of 10 FOVs of two or more independent experiments. The low number of filaments is a result of low binding of the protein construct to actin and do not reflect a lower number of FOVs.

Since the difference in actin curvature is very striking when comparing His6-curly to its truncated counterparts was obvious we did not perform more experiments in the first run (e.g. in case of the Rng2(150-25) construct with linear filaments). We have added some more experiments with the Rng2(1-189)Δ(154-161) construct which confirmed the previous findings. A statistical comparison between the curvatures of the Rng2 truncations with the original Rng2(1-189) using one-way Anova test is included in Figure 2-Figure sup. 1 A. Here we compare the average values of independent experiments (three for each condition, measuring 10 FoV for each).

[Editors' note: further revisions were suggested prior to acceptance, as described below.]

Reviewer #1:1. Based on the data presented in Figure 1 – supplement 1A, the quality of many protein preps is not sufficient for reliable biochemical experiments. This is because His6-Rng2(1-189)del(154-160), His6-Rng2(150-250), and His6-Rng2(41-147) samples appear to contain huge amount of impurities, and are thus not suitable for biochemical experiments. Thus, it is strongly recommended that the authors would either delete all data obtained by using these proteins from the manuscript, or alternatively make much better quality protein preps and repeat the experiments with these.

We appreciate your critical assessment of the biochemical data and agree that it would have been stronger if the protein purity/ stability would have been better. We agree fully with the reviewers that this data is not essential for the central message of the paper and have removed the data from this manuscript. We will reassess the protein expression and add further purification steps (e.g. size exclusion) to repeat the experiments which will form a future publication.

2. The actin filament co-sedimentation assay presented in Figure 1 – supplement 1B and C is not very convincing. I do not understand why the authors carried out this assay with a constant actin concentration and varying the concentration of Rng2(1-189). This is because much more reliable way to carry out such assay would be to keep the concentration of Rng2(1-189) constant, and vary the concentration of actin. This way the possible spontaneous aggregation of Rng2(1-189) would not affect the interpretation of the results. Therefore, these inconclusive data should be deleted from the manuscript. Unless the authors can measure the affinities of various protein constructs for F-actin, they should remove all speculations about different actin-binding sites from the manuscript text (lines 51-54, 63-84, and 195-198).

We are sorry to learn that reviewer#1 did not find the co-sedimentation data convincing. It is true that it could have been better, but we followed the actin-co-sedimentation assay as described in Heier et al., 2017, Jove (https://www.jove.com/t/55613/measuring-protein-binding-to-f-actin-by-co-sedimentation). And the assay was run similarly in (Hayakawa, Y. *et al.,* Actin binding domain of Rng2 strongly inhibits actin movement on myosin II HMM through structural changes of actin filaments. *bioRxiv* (2020) doi:10.1101/2020.04.14.04104) who report an actin binding affinity of ~ 1 μM for the Rng2(1-189) like what we find.

We will remove the data from the paper though and will repeat the co-sedimentation as suggested by the reviewer for future work. This together with studies of the other protein constructs (after optimisation of the purification) will then be more insightful and might form a new publication.

3. I am still not fully convinced about the CP-SNAP647 experiments (presented in Figure 3A and Figure 3 – supplement 1). This is because, in addition to CP-dots at filament ends, also other CP-SNAP647 dots not associated with the filament ends are visible in the images. Moreover, the supplementary video examples are not particularly convincing/high quality. Thus, the authors may also consider removing these data from the manuscript.

We understand that it can be confusing to see non-actin associated fluorescent spots. Unfortunately, it seems that the BG647 fluorophore tends to associate with lipid bilayers. However, these spots can be still distinguished from actin associated dots (that move together with actin) and we did not want to alter the image background to suggest there wouldn’t be any unspecific binding of CP-SNAP647 to the lipid bilayer. Even though there were some actin filaments without CP and some CP without actin, we did not detect any actin filaments with CP that turned the other way, and all CP positive actin filaments, where CP localisation could be clearly associated to an actin filament end, were showing an anti-clockwise orientation. If we would have picked up some random co-localisation of CP and actin, one would expect a more equal distribution between clock- and anti-clockwise orientation.

In addition, our myosin data supports the observation that curly recognises the actin polarisation.

Reviewer #2:(1) I am confused about the y-axis scale on the graph of the amount of filamentous actin each Rng2 construct tethered to the bilayers that the authors included in their response to "Essential Revisions" Question 1. What does it mean for the value of Aactin/Atotal to be 100? Is this graph different from the graph included in Figure 2 Supplement 1 Part B?

A_Actin_/ A_total_ = 100 would mean that the entire field of view is covered by actin. We could have replaced the ‘total’ with ‘FoV’, but the data would be not shown.

(2) I find it curious that the authors did not perform a low-speed co-sedimentation assay with their curly construct to test for filament bundling activity. This would have provided a more quantitative basis for their discussion of whether multiple actin filaments might ever be incorporated into the same ring structure.

Yes, this will be an interesting aspect to look at, and we will do this in future.

(3) In response to my question about a schematic of a myosin II-bound actin ring (originally in Figure 3 Supplement 4 Part E), the authors state that they have made modifications to the model. However, I am unable to locate this schematic in the revised manuscript. Did the authors remove it?

Sorry for this confusion, but we finally decided to not put in any model as we did not have any strong data now to support it. So, we deemed it more sensible to leave it out.

(4) In response to my question about the effects of the counterclockwise curvature on the structure of the actin filaments, the authors state: "However, data by XXX indicates that binding of curly can loosen/disrupt the helical structure of actin filaments…". Who is XXX?

Our apologies for this omission, the paper referred to is Hayakawa, Y. et al., Actin binding domain of Rng2 strongly inhibits actin movement on myosin II HMM through structural changes of actin filaments. bioRxiv (2020) doi:10.1101/2020.04.14.04104

(5) In response to my question about binding/structural cooperativity, the authors state that they do not have direct evidence to determine whether binding of curly to actin filaments is cooperative, or if filament curvature arises from an additive effect of multiple independent binding events. To reflect this fact, they indicate that they have rephrased the following (original) statement using more cautious language:"Importantly, the uni-directional bending supports the hypothesis that the binding site of curly with actin filaments defines an orientation, and the propagation of a curved trajectory once established indicates a cooperative process."However, their revised statement still appears to suggest a cooperative model for binding/bending without acknowledging the possibility of other binding models.

Our apologies, the statement was now altered by removing the mention of a cooperative process.

“Importantly, the uni-directional bending supports the hypothesis that the binding site of curly with actin filaments defines an orientation with the propagation of an established curved trajectory.”

(6) The authors mention that they have added a quantification of ring frequency in curly-expressing cells compared to cells expressing only Lifeact-mCherry. Perhaps I have misunderstood their graphs, but I was unable to locate this analysis (or any micrographs or analysis with either wild-type, "EGFP alone" or "Lifeact-mCherry alone" cells).

A box plot showing the number of rings per cell at the basal site for HEK293 cells transfected with EGFP-curly, EGFP-curly-CAAX and RPE-1 with EGFP-Curly was shown in Figure 4 – supplement figure 2. We have added now also the data for LifeAct-mCherry expressing cells that we forgot to add in the previous submission.

Reviewer #3:The authors have addressed all my previous concerns (reviewer 3 in the rebuttal).However, I think the authors have left a place holder in their response to reviewer 2 (However, data by XXX indicates that binding of curly can loosen/ disrupt the helical structure of actin filaments leading to kinks. This would indicate towards the 2nd suggested mechanisms), as I do not know who XXX is meant to indicate.

Our apologies for this omission, the paper referred to is Hayakawa, Y. et al., Actin binding domain of Rng2 strongly inhibits actin movement on myosin II HMM through structural changes of actin filaments. bioRxiv (2020) doi:10.1101/2020.04.14.04104